# Cancer-Associated Thrombosis on Bevacizumab: Risk of Recurrence and Bleeding When Bevacizumab Is Stopped or Continued

**DOI:** 10.3390/cancers15153893

**Published:** 2023-07-31

**Authors:** Marie Mayenga, Nicolas Falvo, Isabelle Mahé, Anne-Sophie Jannot, Benoit Gazeau, Guy Meyer, Nicolas Gendron, Olivier Sanchez, Sadji Djennaoui, Benjamin Planquette

**Affiliations:** 1Department of Pulmonology and Intensive Care, Hôpital Européen Georges-Pompidou, AP-HP, 75015 Paris, Francebenjamin.planquette@aphp.fr (B.P.); 2Department of Vascular Pathology, Centre Hospitalier Universitaire Dijon-Bourgogne, 21000 Dijon, France; 3Université Paris Cité, Service de Médecine Interne, Hôpital Louis Mourier, AP-HP, 92700 Colombes, France; 4Innovative Therapies in Haemostasis, INSERM UMR_S1140, 75006 Paris, France; 5INNOVTE-FCRIN, 42055 Saint-Etienne, France; 6Department of Biostatistics, Medical Informatics and Public Health, Hôpital Européen Georges-Pompidou, AP-HP, 75015 Paris, France; 7Department of Respiratory Medicine, Centre Hospitalier de Bourg-en-Bresse, 01012 Bourg-en-Bresse, France; 8Université Paris Cité, Innovative Therapies in Haemostasis, INSERM, 75006 Paris, France; 9Department of Biological Hematology, Hôpital Européen Georges-Pompidou, AP-HP, 75015 Paris, France; 10Université de Paris, Innovative Therapies in Haemostasis, Laboratoire de Recherches Biochirugicales (Fondation Carpentier), 75005 Paris, France

**Keywords:** anti-angiogenic drug, anticoagulant, bevacizumab, bleeding, cancer, recurrence, venous thrombosis

## Abstract

**Simple Summary:**

Cancer-associated thrombosis is a frequent complication and a poor prognostic event. Bevacizumab is an antiangiogenic drug used in the treatment of many cancers. Few data are available on the concomitant use of bevacizumab and anticoagulant therapy. The aim of this retrospective multicenter study was to determine the safety and efficacy of anticoagulant therapy in patients receiving bevacizumab. We observed that patients with a cancer-associated thrombosis on bevacizumab did not experience more bleeding complications or thromboembolic recurrences on anticoagulant therapy if they continued bevacizumab. The safety and efficacy of anticoagulant therapy did not appear to be affected by bevacizumab, and these results encourage clinicians to continue this drug.

**Abstract:**

Cancer-associated thrombosis (CAT) is a common complication during cancer, with complex management due to an increased risk of both recurrence and bleeding. Bevacizumab is an effective anti-angiogenic treatment but increases the risk of bleeding and potentially the risk of venous thromboembolism (VTE). The aim of this study was to evaluate the efficacy and safety of anticoagulant therapy in patients with CAT receiving bevacizumab, according to the continuation or discontinuation of bevacizumab. In a retrospective multicenter study, patients receiving anticoagulant for CAT occurring under bevacizumab therapy were included. The primary endpoint combined recurrent VTE and/or major or clinically relevant non-major bleeding. Among the 162 patients included, bevacizumab was discontinued in 70 (43.2%) patients and continued in 92 (56.8%) patients. After a median follow-up of 318 days, 21 (30.0%) patients in the discontinuation group experienced VTE recurrence or major or clinically relevant non-major bleeding, compared to 27 (29.3%) in the continuation group. The analysis of survival following the first event showed no significant difference between the groups in uni- or multivariate analysis (*p* = 0.19). The primary endpoint was not influenced by the duration of bevacizumab exposure. These results suggest that the efficacy and safety of anticoagulant therapy in patients with CAT receiving bevacizumab is not modified regardless of whether bevacizumab is continued or discontinued.

## 1. Introduction

The risk of venous thromboembolism (VTE) is 4 to 7 times higher in patients with cancer compared to the general population [1,2]. The annual incidence rate depends on factors such as time to diagnosis, cancer characteristics, anticancer treatments, and patient comorbidities [2,3,4] Cancer-associated thrombosis (CAT) is the second leading cause of death in patients with cancer. This population presents a higher risk of mortality and VTE recurrence than the general population [5,6], along with a two-fold higher risk of bleeding during treatment [6,7,8]. The three months following the diagnosis of CAT are at highest risk for pulmonary embolism recurrence and fatal bleeding [6].

Bevacizumab is a humanized monoclonal antibody specifically directed against vascular endothelial growth factor A (VEGF-A). This anti-angiogenic drug, approved beginning in 2005, has shown efficacy in combination with chemotherapy, with an impact on progression-free survival and overall survival (OS) [9,10,11,12]. Bevacizumab is now part of the therapeutic arsenal for colorectal, lung, and ovarian cancers as well as glioblastoma. It has been reported that bevacizumab increases the risk of bleeding by two to three times [13,14] as well as the risk of major bleeding, particularly in colorectal, renal, and lung cancers [15]. Furthermore, the risk of CAT under bevacizumab remains uncertain, with conflicting results in meta-analyses [13,16,17,18].

Nonetheless, the occurrence of a thromboembolic event in patients receiving bevacizumab is a frequent situation. In practice, discontinuing bevacizumab would constitute the loss of an effective anticancer treatment; however, maintaining it could expose patients to thrombotic and/or bleeding complications. Few data are available to evaluate the efficacy and tolerance of anticoagulant therapy associated with bevacizumab [19].

The objective of the present study was to determine whether continuing bevacizumab in patients with CAT is associated with an increased risk of thromboembolic recurrence and bleeding under anticoagulant treatment.

## 2. Materials and Methods

### 2.1. Study Design

The present study was a multicenter retrospective cohort study conducted in the Georges Pompidou European Hospital (HEGP, APHP, Paris, France), the Louis Mourier Hospital (APHP, Colombes, France), and the Georges François Leclerc Cancer Center (Dijon, France).

This study was registered with the French Data Protection Authority (CNIL, registration number 2223564) and was performed in accordance with the published reference methodology MR-004, adapted to the existing framework of the EU General Data Protection Regulation (GDPR). The MR-004 concerns non-human research, health studies or evaluations, and research that only reuses data that have already been collected. Therefore, in this study all retrospective health data were de-identified before analysis.

### 2.2. Patients, Inclusion Criteria, and Non-Inclusion Criteria

Eligible patients were consecutive subjects over 18 years of age with histologically proven solid cancer treated with bevacizumab and who presented objectively confirmed deep vein thrombosis (DVT) and/or pulmonary embolism (PE) according to international guidelines [20] that occurred during bevacizumab therapy. VTE included DVT defined as sural, popliteal, femoral, iliac, vena cava, jugular, right atrial, and port-a-cath associated thrombosis. Patients had to be receiving anticoagulant therapy at therapeutic dose (low molecular weight heparin, LMWH, fondaparinux, direct oral anticoagulant, DOACs) for VTE.

Exclusion criteria included preexisting anticoagulation, whatever the dosage, contraindication to anticoagulant therapy, and contraindication to vitamin K antagonists. Visceral and upper extremity vein thrombosis unrelated to the presence of a port-a-cath were not included.

Patients were identified using the computer software of each center (see Appendix A). Demographic and oncologic data were collected at the time of cancer and VTE diagnosis. CAT type, date of occurrence, localization, and response to oncological treatment at the time of CAT were collected, as well as the type, date of initiation, and discontinuation of the anticoagulant treatment.

### 2.3. Surveillance and Follow-up

The follow-up period started at the time of CAT diagnosis, with a censor date of 07/31/2021. The number of days of bevacizumab exposure after CAT diagnosis was collected for each patient. A patient was considered as exposed for up to 30 days after each bevacizumab injection, taking into account the half-life of bevacizumab, which is 18 days in women and 20 days in men, along with its decay [21].

Two groups of patients were defined: patients in whom bevacizumab was discontinued after CAT, and patients in whom bevacizumab was continued. Bevacizumab discontinuation was defined as no new injection of bevacizumab or discontinuation of treatment for at least 3 months after CAT diagnosis.

### 2.4. Endpoints

The primary endpoint combined VTE recurrence or major or clinically relevant non-major bleeding (CRNMB) throughout the follow-up period. Secondary endpoints were the occurrence of VTE recurrence, major bleeding and clinically relevant non-major bleeding throughout follow-up, and overall survival.

Thromboembolic recurrence was defined as the occurrence of a new VTE event in a vascular territory different from the initial one or the extension of the initial thrombus. The recurrence had to be objectively confirmed by an imaging technique (ultrasound or CT venography for DVT and port-a-cath associated thrombosis; computed tomography pulmonary angiogram or pulmonary ventilation-perfusion lung scan for PE).

Major bleeding and CRNMB were defined according to the International Society on Thrombosis and Haemostasis (ISTH) criteria [22]. Major bleeding included bleeding resulting in death, symptomatic major organ bleeding (intracranial, intraspinal, intraocular, retroperitoneal, intra-articular pericardial, or intramuscular due to compartment syndrome), and symptomatic bleeding resulting in at least a 20 g/L decrease in hemoglobin or requiring transfusion of two or more units of whole blood or red cells. CRNMB included non-major bleeding requiring medical or surgical intervention. Other bleeding was considered minor and was not included in the analysis.

### 2.5. Statistical Analysis

Results were expressed as medians (interquartile range, IQR, 5th–95th percentiles).

Recurrence-free and bleeding-free survival based on stopping or continuing bevacizumab was determined by the Kaplan–Meier method. Only the first event was considered in case of multiple events, including for secondary endpoint analyses. Comparison of survival between groups was performed by the log-Rank test. Survival was analyzed by a univariate and then multivariate Cox model. The multivariate model considered the baseline characteristics of the population with a statistically different distribution between the bevacizumab discontinuation group and the continuation group. The difference between the groups was considered significant at a *p* < 0.05.

A sensitivity analysis was performed by taking exposure as a quantitative variable, i.e., considering the duration of bevacizumab exposure after the first VTE event. The start and end of bevacizumab treatment periods were recorded for each patient, and bevacizumab exposure was inferred considering that a patient was exposed for 30 days after a bevacizumab injection. The time to the first event (major bleeding or CRNMB or VTE recurrence), if any, or to death, if any, was studied in a Cox model with bevacizumab exposure as the explanatory variable, considering bevacizumab exposure as a time-dependent variable. Data were censored on 31 July 2021.

## 3. Results

### 3.1. Population Characteristics

Between June 2006 and December 2020, a total of 1543 patients were identified in the three hospitals: 932 patients from the Georges François Leclerc Center, 415 patients from the Georges Pompidou European Hospital, and 196 patients from the Louis Mourier Hospital. Among them, 162 patients were finally included for presenting a CAT during bevacizumab treatment. Among these, bevacizumab was continued in 92 patients (continuation group) and discontinued in 70 patients (discontinuation group) (Figure 1).

The characteristics of the population are described in Table 1. The median age was 64 years (IQR 55–71), there was a majority of women (59%), and most of the patients had a performance status between 0 and 1. There was no difference between the two groups regarding baseline demographic characteristics. Colorectal cancer was the most frequent cancer (Appendix A). There was no difference in the distribution of cancer types between the groups; however, metastatic status was significantly more frequent in the continuation group (*p* = 0.02). Progression was significantly more represented in the group who stopped bevacizumab (*p* = 0.0001).

Table 2 displays baseline VTE events among patient groups. The index event was a DVT in 58 (36%) patients, PE in 81 (50%) patients, and the combination of both DVT and PE in 23 (14%) patients. The median time between bevacizumab initiation and VTE diagnosis was 79 days (95% CI 39–154). The median time from first inclusion of the study (June 2006) to CAT diagnosis was significantly higher in the bevacizumab continuation group (*p* = 0.03), with more bevacizumab discontinuation in the initial period of the study (61% before 2011 vs. 37% after 2011). Most patients were treated with LMWH, and only five (3%) patients received DOACs. Discontinuation of anticoagulant therapy was reported in 52 (36%) patients, with no difference between the discontinuation and continuation groups (*p* = 0.23).

### 3.2. Primary Endpoint

The median follow-up was 318 days (IQR 128–779). During follow-up, the primary endpoint occurred in 48 patients (30%): 27 patients (29%) in the continuation group and 21 patients (30%) in the discontinuation group (Table 3).

In the whole population, the presence of renal failure, the use of antiplatelet therapy, and the type of anticoagulation did not influence the occurrence of the primary endpoint or the location of the primary cancer. Similarly, the presence of secondary cerebral or hepatic locations did not increase the risk of hemorrhage or recurrence (Appendix A).

In the survival analysis, no significant difference was found between groups for the occurrence of the primary endpoint (hazard ratio 0.67 for continuation, 95% CI (0.38–1.24), *p* = 0.19) (Figure 2). Considering metastatic disease, tumor progression, and time from first study inclusion to CAT, there was no significant difference for the primary endpoint between stopping and continuing bevacizumab (HR 1.33 for stopping, 95% CI (0.68–2.61), *p* = 0.40).

Patients with tumor progression were more prevalent in the bevacizumab discontinuation group. A subgroup analysis was performed in 116 patients without progression and showed no difference for the occurrence of the primary endpoint between patients who stopped or continued bevacizumab (hazard ratio, HR 0.76 for continuation, 95% CI 0.38–1.51, *p* = 0.36) (Appendix A).

Median bevacizumab exposure after CAT was 187 days in the continuation group and 20 days in the discontinuation group. The cumulative number of days on bevacizumab was 24 days in the continuation group and 7 days in the interruption group. There was no difference in the occurrence of the primary end point based on bevacizumab exposure or not, with an HR for exposure of 0.69 (95% CI 0.34–1.3, *p* = 0.28).

### 3.3. Bleedings

Bleeding occurred in 27 patients (17%), major bleeding in 10 patients (6%), and CRNMB in 17 patients (10%). Major bleeding included five (50%) gastrointestinal hemorrhages (all in patients with colorectal cancer), one retroperitoneal hematoma, one psoas hematoma, one skin hemorrhage, and two brain hemorrhages of metastatic lesions. Four major bleedings resulted in death. As one of these deaths was preceded by a recurrence of VTE, only the other three deaths were included in the primary endpoint. Two of the fatal hemorrhages were secondary to cerebral hemorrhages (continuation group) and one was due to hemorrhagic shock on rectorrhagia (discontinuation group). CRNMB was mainly gastrointestinal bleeds (n = 7), followed by epistaxis (n = 5) and hematoma (n = 5).

There was no difference in the occurrence of bleeding complications depending on whether bevacizumab was stopped or continued (Appendix A) (HR 0.64 for continuation, 95% CI 0.29–1.38, *p* = 0.23). There was no difference in the occurrence of major bleeding whether bevacizumab was discontinued or not (HR 0.83 for continuation, 95% CI 0.23–3.02, *p* = 0.77, Appendix A). Most patients with bleeding (92%) were still on anticoagulant therapy at the time of the event, which was statistically different from the non-bleeding population (37%, *p* = 0.002). Anticoagulant therapy was then discontinued in all seven patients surviving the major bleeding event. Platelet counts at the time of bleeding were available for 24 of the 27 patients. Among them, seven patients had platelet counts below 150 G/L, including four patients with platelet counts below 100 G/L.

### 3.4. Recurrences

Twenty-one (13%) recurrences occurred: six PE (4%), 13 DVT (8%), and two combinations of both PE and DVT without fatal outcome (1%). Nine patients (43%) recurred with the same presentation as the initial episode. In patients with recurrence, discontinuation of anticoagulant therapy was non-significantly more frequent than in patients without recurrence (n = 10, 46% vs. n = 40, 32%, *p* = 0.22). There was no difference in the occurrence of recurrence depending on whether bevacizumab was stopped or continued (HR 0.93 for continuation, 95% CI 0.40–2.19, *p* = 0.87) (Appendix A). Fifty-four per cent of patients with a VTE recurrence had stopped anticoagulant therapy.

### 3.5. Overall Survival

During follow-up, 139 (90%) patients died, 80 in the continuation group (87%) and 59 in the discontinuation group (82%). There was no difference in OS depending on whether bevacizumab was stopped or continued (HR 0.82 for continuation, 95% CI 0.58–1.15, *p* = 0.23) (Figure 3). Survival analysis showed a significantly higher OS rate at 6 months in the continuation group compared to the interruption group (79% vs. 59%, *p* = 0.005), and similarly for the OS rate at 12 months (60% vs. 41%, *p* = 0.03). This benefit was no longer observed at 2 years, with OS rates of 13% and 16% in the continuation and discontinuation groups, respectively (*p* = 0.66), nor at 5 years (OS of 7% and 13%, respectively, *p* = 0.29).

Deaths were related to cancer in most cases (n = 115, 83%), with no difference between the two groups; 81% of deaths were related to cancer in the continuation group and 85% in the discontinuation group (*p* = 0.38) (Table 4).

## 4. Discussion

In patients presenting a CAT during bevacizumab treatment, this retrospective multicenter study found no difference in VTE recurrence or major bleeding and CRNMB between the group discontinuing or continuing bevacizumab. No difference in recurrence, bleeding, or major bleeding considered separately was observed throughout the follow-up period. Finally, no difference in overall survival between the two groups was observed over the entire follow-up period, although there was a lower mortality rate in the continuation group within one year of CAT. Thus, there was no difference in safety or efficacy of anticoagulant therapy between patients exposed and unexposed to bevacizumab. This result suggests that bevacizumab could be continued in patient anticoagulated for CAT.

Several meta-analyses have examined the risk of VTE on bevacizumab, with conflicting results [13,16,17,18,23]. In practice, patients treated with bevacizumab are at high risk of CAT because of their cancer type and metastatic status. Therefore, the clinician is challenged with the question of anticoagulation and maintenance of bevacizumab in patients receiving anticoagulation at therapeutic doses after CAT occurrence. Few data are available on the risk of bleeding and recurrence of CAT on concomitant anticoagulants at therapeutic dose and bevacizumab. The most recent review of anti-angiogenic agents does not cover this aspect of management [24]. Therefore, in the present study we chose to specifically address this clinical question by assessing the risk of VTE recurrence and bleeding on anticoagulation based on continuation or discontinuation of bevacizumab.

Bleedings occurred in 17% of our patients for a median follow-up of 318 days. This relatively high rate may be explained by the characteristics of the population, which was composed of 84% patients with metastatic cancers, 37% platinum salts-treated patients, and nearly half patients with colorectal cancers who were at high risk of bleeding [25]. In comparison, a recent meta-analysis found 15% to 18% with bleeding at one year in patients treated by LMWH for a CAT [26]. In the dalteparin arm of the CLOT trial the bleeding rate was 14% at 6 months [27], compared to 10% at 6 months in the dalteparin arm of the CARAVAGGIO trial [28] and 15% at one year in the dalteparin arm of the HOKUSAI-VTE trial [29]. The 13% recurrence rate in our patients was comparable to those reported in the literature. In comparison, in the dalteparin arms the recurrence rates were 9% at 6 months in the CLOT trial [27], 8% at 6 months in the CARAVAGGIO trial [28], and 11% at one year in the HOKUSAI-VTE trial [6,29,30,31].

Exposure was defined as a binary variable according to the meta-analysis by Scappaticci et al. [17]. This definition does not incorporate exposure durations. In the literature, only a few meta-analyses have incorporated duration of exposure into the analysis of the incidence of CAT on bevacizumab, leading to conflicting results [13,16] which might be explained by the difference in exposure definition. Therefore, we performed a sensitivity analysis based on exposure duration, which showed no association with CAT incidence. The measure of the total duration of bevacizumab exposure for each patient confirmed the consistency of the binary group definition, with a significant difference between the two groups.

Most patients in our study were treated with LMWH. Following the CARAVAGGIO trial [28], DOACs have become increasingly important in the management of CAT, and most patients in this study would likely have access to apixaban therapy today. Our study cannot conclude on their use, because only five of the patients were initially managed with DOACs. Detailed data on recurrence of CAT and bleeding in the 59 patients treated with bevacizumab in the CARAVAGGIO trial are not available [28]. In the HOKUSAI-VTE trial, 19 patients received bevacizumab at baseline in the edoxaban arm, 30 in the dalteparin arm, and 13 and 19 patients, respectively, continued treatment after the diagnosis of CAT [29]. While the safety of edoxaban was identical to that of dalteparin, interpretation is limited by the small number of patients. The rates observed in patients treated with bevacizumab were comparable to the rates observed in our population and higher than in the overall trial population [29].

In the present study, the patients presenting bleeding during the follow-up period received longer duration of anticoagulant therapy, with 92% receiving anticoagulation at the time of bleeding, and the risk of bleeding seems to be primarily related to anticoagulant therapy rather than to bevacizumab exposure. This conclusion is consistent with a study that assessed the bleeding risk of anticoagulant therapy associated with bevacizumab versus chemotherapy in three clinical trials: major and nonmajor bleeding rates were not higher in patients on bevacizumab [19].

Finally, our study in patients with CAT that occurred on bevacizumab shows that the benefit/risk ratio of anticoagulant therapy does not appear to be altered by continuation or discontinuation of antiangiogenic therapy. The practices seem to have changed over time; bevacizumab was less frequently discontinued after 2011, and discontinuation was more frequently related to patients with progression, suggesting that bevacizumab discontinuation was not justified by a higher bleeding risk.

This work has several limitations. On the one hand, considering the importance of the role of exposure to anticoagulant therapy on the occurrence of the primary endpoint, the impossibility of assessing the precise duration of exposure and dosage in each group represents one of the limitations of this retrospective work. However, the type of anticoagulation at initiation was available for all patients, and the discontinuation or not of anticoagulation therapy was available for 145 patients. On the other hand, the reason for bevacizumab discontinuation may be a confounding factor in two situations. First, when bevacizumab was stopped because of oncological progression, i.e., in patients at risk for CAT recurrence; a subgroup analysis of progression-free patients showed similar results and accounted for this bias. Second, when bevacizumab discontinuation was motivated by the CAT occurrence itself; the clinician’s choice might have been influenced by the patient’s bleeding risk, and bevacizumab would be more likely to be discontinued in patients at higher risk. However, the major risk factors for bleeding were not different between the discontinuation and continuation groups.

Finally, this was a retrospective study with a relatively small number of patients, which did not allow us to make a non-inferiority hypothesis. Confirmation of the safety and efficacy of anticoagulation associated with bevacizumab therapy would require a prospective non-inferiority study; however, a randomized discontinuation of bevacizumab does not seem feasible for ethical reasons considering these preliminary results, particularly those on survival. As an alternative, a descriptive cohort study could be envisaged.

## 5. Conclusions

Today, bevacizumab is a targeted therapy that plays a central role in the first-line management of many cancers. In this multicenter retrospective study, patients with CAT on bevacizumab did not experience more bleeding complications or thromboembolic recurrence on anticoagulants if they continued bevacizumab. Thus, the safety and efficacy of anticoagulant therapy do not appear to be affected by bevacizumab, and these results encourage its continuation. A larger study seems necessary to confirm these results and to definitively establish this strategy of continuation of antiangiogenic therapy.

## Figures and Tables

**Figure 1 cancers-15-03893-f001:**
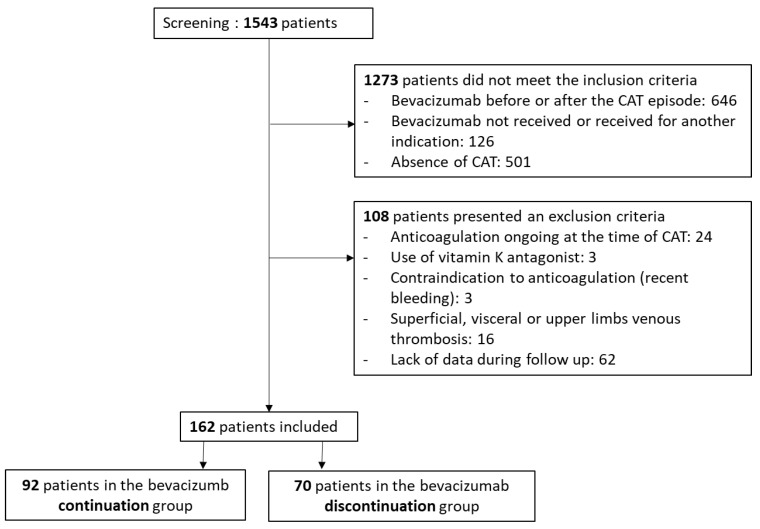
Flow chart.

**Figure 2 cancers-15-03893-f002:**
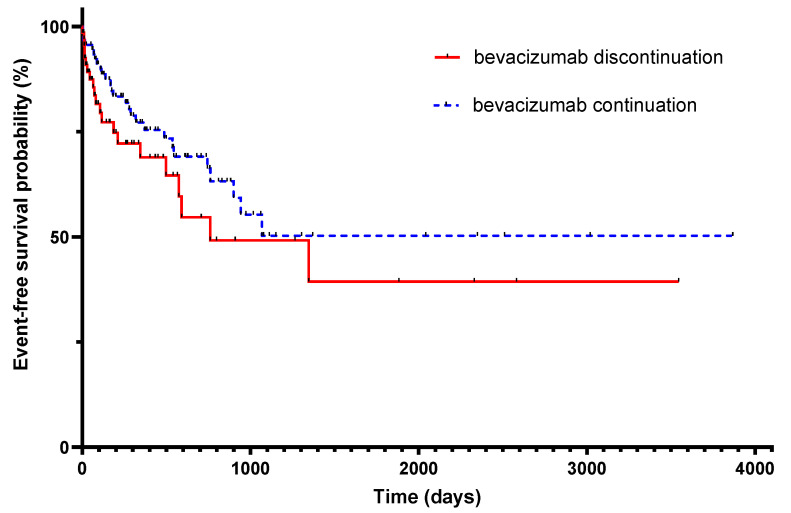
Occurrence of recurrent cancer-associated thrombosis or bleeding according to the continuation or discontinuation of bevacizumab. Hazard Ratio 0.67 for continuation, 95% CI (0.38–1.24), *p* = 0.19.

**Figure 3 cancers-15-03893-f003:**
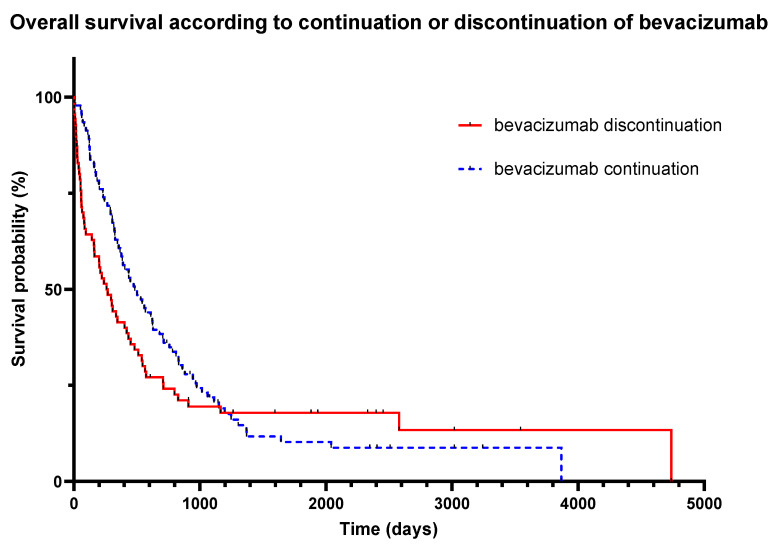
Overall survival according to continuation or discontinuation of bevacizumab. Hazard ratio 0.82 for continuation, 95% CI (0.580–1.151), *p* = 0.23.

**Table 1 cancers-15-03893-t001:** Baseline demographic and oncologic characteristics of the population. Variables are expressed as median (interquartile range) or absolute value (percentage).

	Population n = 162	Bevacizumab Continuation Group, n = 92	Bevacizumab Discontinuation Group, n = 70	*p*-Value
Age at cancer diagnosis	64 [55–71]	64 [54–71]	64 [57–1.5]	0.82
Sex				
Female	96 (59%)	58 (63%)	38 (54%)	0.33
Male	66 (41%)	34 (37%)	32 (46%)	
BMI (n = 156)	24.6 [21–28]	24 [20–27]	24.6 [22–28]	0.24
Performance Status				
0–1	130/153 (85%)	75/85 (88%)	55/68 (81%)	0.18
2–3	23 (15%)	10 (12%)	13 (19%)	0.26
Renal failure (eGFR < 60 mL/mn)	27/144 (19%)	15/92 (16%)	12/63 (19%)	>0.99
Anemia (Hb < 100 g/L)	11/153 5 (7%)	4/86 (5%)	7/67 (10%)	0.21
Thrombocytosis (>300 G/L)	38/148 (26%)	19/85 (22%)	19/63 (30%)	0.34
Antiaplatelet therapy	17 (11%)	11 (12%)	6 (9%)	0.61
Cancer type				
Colorectal	78 (48%)	39 (42%)	39 (56%)	0.11
Ovarian, endometrial	28 (17%)	20 (21%)	8 (11%)	0.10
Breast	15 (9%)	10 (11%)	5 (7%)	0.59
Lung	18 (11%)	13 (14%)	5 (7%)	0.21
Central nervous system	17 (11%)	7 (8%)	10 (14%)	0.20
Others	6 (4%)	3 (3%)	3 (4%)	0.99
Histological subtype				
Adénocarcinome	119 (73%)	67 (73%)	52 (74%)	0.86
Current treatment line				
1st line	74/152 (49%)	44/85 (52%)	30/67 (45%)	0.43
2nd line	50 (33%)	24 (28%)	26 (39%)	0.22
3rd line	28 (18%)	17 (20%)	11 (16%)	0.68
Platinum salt treatment	62 (38%)	36 (39%)	26 (37%)	0.87
Metastatic disease	123/146 (84%)	77/85 (91%)	46/61 (75%)	0.02
Metastases				
Cerebral	9 (6%)	6 (6%)	3 (4%)	0.73
Hepatic	57 (35%)	29 (32%)	28 (40%)	0.32

BMI: body mass index, eGFR: estimated glomerular filtration rate, Hb: hemoglobin.

**Table 2 cancers-15-03893-t002:** Initial thromboembolism characteristics. Variables are expressed as median (interquartile range), or absolute value (percentage).

	Population n = 162	Bevacizumab Continuation Group, n = 92	Bevacizumab Discontinuation Group, n = 70	*p*-Value
CAT localization				
DVT	58 (36%)	34 (37%)	24 (34%)	0.74
PE	81 (50%)	48 (52%)	33 (47%)	0.63
DVT and PE	23 (14%)	10 (11%)	13 (19%)	0.18
Most proximal obstruction of PE				
Segmental or more proximal	79/97 (81%)	46/54 (85%)	33/43 (76%)	0.31
Subsegmental	18 (19%)	8 (15%)	10 (23%)	
Unilateral	55/98 (56%)	32/56 (57%)	23/42 (55%)	0.84
Bilateral	43 (44%)	24 (43%)	19 (45%)	
Discovery mode Clinically suspected	82/160 (51%)	44/91 (48%)	38/69 (55%)	0.43
Incidental asymptomatic	65 (41%)	42 (46%)	23 (33%)	0.11
Incidental symptomatic	13 (8%)	5 (6%)	8 (12%)	0.24
Time between CAT and first inclusion (days)	3080 [1947–4089]	3239 [2931–3554]	2528 [2313–3505]	0.03
LMWH therapy	152 (94%)	87(95%)	65 (93%)	0.75
LMWH discontinuation during the follow-up	52/145 (36%)	31/79(39%)	21/66 (32%)	0.23
Time between bevacizumab initiation and CAT (days)	79 [39–154]	77 [41–141]	84 [35–173]	0.55
Bevacizumab posology at CAT diagnosis (mg/kg) (n = 135)	7.5 [5–10]	7.5 [5–10]	5 [5–10]	0.20
Other risk factor of CAT	37 (23%)	18 (20%)	19 (21%)	0.26
Response to oncologic treatment Response	40/154 (26%)	27/87 (31%)	13/67 (19%)	0.14
Stability	76 (49%)	50 (57%)	26 (38%)	0.02
Progression	38 (25%)	10 (11%)	28 (42%)	0.0001

CAT: cancer associated thrombosis, DVT: deep venous thrombosis, PE: pulmonary embolism, LMWH: low molecular weight heparin.

**Table 3 cancers-15-03893-t003:** Occurrence the primary endpoint (CAT recurrence and bleeding) during follow-up. Variables are expressed as absolute value (percentage).

	Population n = 162	Bevacizumab Continuation Group, n = 92	Bevacizumab Discontinuation Group, n = 70
CAT recurrence or bleeding	48 (30%)	27 (29%)	21 (30%)
Recurrence	21 (13%)	13 (14%)	8 (11%)
Bleeding	27 (17%)	14 (15%)	13 (19%)
Major bleeding	10 (6%)	6 (7%)	4 (6%)
Clinically relevant non major bleeding	17 (11%)	8 (8%)	9 (13%)

**Table 4 cancers-15-03893-t004:** Deaths and causes of death in the study population. Variables are expressed as absolute value (percentage).

	Population n = 162	Bevacizumab Continuation Group, n = 92	Bevacizumab Discontinuation Group, n = 70
Death	139 (90%)	80 (87%)	59 (82%)
Cancer related deaths	115 (83%)	65 (81%)	50 (85%)
Fatal bleeding	4 (3%)	3 (4%)	1 (2%)
Other cause of death	12 (9%)	6 (8%)	6 (10%)
Unknown cause of death	4 (3%)	6 (8%)	2 (3%)

## Data Availability

The data presented in this study are available on request from the corresponding author. The data are not accessible to the public for reasons of medical confidentiality.

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
