# Peer review of "Cancer-Associated Thrombosis on Bevacizumab: Risk of Recurrence and Bleeding When Bevacizumab Is Stopped or Continued"

_cancers, 2023, doi:10.3390/cancers15153893_

Round 1

Reviewer 1 Report

Mayenga, Planquette et al present a muticenter retrospective cohort study of recurrent cancer-associated venous thrombosis (CAT) after continuation or cessation of the anti-angiogenic monoclonal antibody bevacizumab.  The study evaluated whether continuation of bevacizumab therapy after sentinel CAT was associated with increased CAT recurrence or bleeding as compared to discontinuation of bevacizumab after sentinel CAT.  Justification for this concept was that bevacizumab is now incorporated into many solid tumor treatment regimens and meta-analyses of CAT outcomes while on bevacizumab have been conflicting.  The authors screen 1543 patients on bevacizumab in a French hospital system and ultimately included 162 patients for this retrospective analysis: 92 individuals continued bevacizumab after CAT and 70 discontinued bevacizumab.  Importantly, the authors found that 30% of individuals had CAT recurrence or bleeding whether or not they continued bevacizumab and that neither major nor total bleeding was increased by continuing bevacizumab treatment.  

The study was well-written and easy to follow.  The definition of CAT was clear.  Anticoagulation must have been therapeutic doses.  Bevacizumab discontinuation and exposure were clearly defined.  CAT recurrence was well-defined, and took into account that initial clot may not fully resolve.  Importantly, major bleeding and clinically-relevant non-major bleeding were defined according to an international society standard.  

Numbers identified in the tables are consistent throughout the text.  

The discussion was robust and addressed questions that arose to me while reviewing the results section.  The authors directly address prior published conflicting results, varying anticoagulants, the high rate of bleeding observed.  Study limitations were also well-identified, including "precise duration of exposure and dosing" of anticoagulants in each group.  As well, the discussion included what appears to be a pivotal change in practice in 2011 of largely continuing bevacizumab after CAT (possibly due to publication that year of a meta-analysis showing no difference in thrombo-hemorrhagic outcomes whether bevacizumab was continued or not after CAT).  

In summary, this is a very well-written and well-designed study that adds to our current understanding of the safety of ongoing bevacizumab treatment after CAT.  

The only minor issue I found was on line 254 "TVP" is reported, but I do not see "TVP" defined: perhaps this is a typographic error?

Author Response

We thank you for your review and your interesting comments. 

You have reported the presence of the term "TVP" on line 254. Thank you for pointing this out. This is a typographic error - it's the term "DVT" - we apologize and have made the change in the text.

Yours faithfully

Reviewer 2 Report

The authors of this paper have put a significant effort to thoroughly assess the role of bevacizumab in the cancer-associated thrombosis setting. The facts are clearly presented and explained and the question that initiated this study is adequately answered.  

Author Response

We thank you for your review and your comments. 

Yours Faithfully